# Experimental and Numerical Study on a Non-Explosive Reactive Armour with the Rubber Interlayer Applied against Kinetic-Energy Penetrators—The ‘Bulging Effect’ Analysis

**DOI:** 10.3390/ma14123334

**Published:** 2021-06-16

**Authors:** Teresa Fras

**Affiliations:** French-German Research Institute of Saint-Louis (ISL), 68301 Saint-Louis, France; teresa.fras@isl.eu

**Keywords:** passive non-reactive armour, NERA, natural rubber, DOP, bulging effect, kinetic-energy penetrator (KEP), long-rod projectiles

## Abstract

The study concerns a protection system applied against kinetic-energy penetrators (KEPs) composed of steel plates sandwiching a rubber layer. Laminated steel-elastomer armours represent non-explosive reactive (NERA) armours that take advantage of a so-called ‘bulging effect’ to mitigate KEP projectiles. Upon an impact, the side steel plates deform together with the deforming rubber interlayer. Their sudden deformation (bulging) in opposite directions disturbs long and slender KEP projectiles, causing their fragmentation. The presented discussion is based on the experimental investigation, confirming that the long-rod projectiles tend to fracture into several pieces due to the armour perforation. A numerical simulation accompanies the ballistic test providing an insight into the threat/target interactions. The presented experimental–numerical study explains the principles of the analysed protection mechanism and proves the efficiency of the materials composition making up the laminated non-reactive protection system.

## 1. Introduction

Modern protection systems aimed to mitigate kinetic-energy penetrators (KEP, KE projectiles, and long-rod projectiles) are based on an optimisation of mechanisms due to which the threats are decelerated, fragmented, deflected or strongly eroded before hitting the main armour. The KE projectiles are an ammunition type designed to penetrate thick armours of battle tanks or heavy armoured vehicles [1]. Their penetration potential is increased by minimising the projectile’s diameter and maximising its length and mass by applying the densest metals, i.e., depleted uranium or tungsten heavy alloys (WHA). A KEP projectile, accelerated to the impact velocities higher than 1000 m/s, brings a large amount of the kinetic energy upon a minimal area of the target, burrowing a deep cavity through its structure. Typically, to avoid thick and heavy layers of homogenous armours, active or reactive protection solutions are applied against long-rod projectiles [2].

In the reactive and active systems, a part of the armour is propelled against the incoming threat either at some distance away from a protected combat vehicle (active) or at the moment of an impact (reactive). Thus, in active protection systems, a plate/projectile is launched towards an incoming threat, whereas reactive protection systems reuse the projectile’s impact energy. The explosive reactive armours (ERA) consist of a high-energy explosive interlayer, which explodes, accelerating the front plate towards a kinetic threat at the moment of impact. As a result, a projectile may be destabilised and strongly fragmented. Instead of explosive material, the steel plates sandwich a material (elastomers, polymers), which causes a rapid deformation (bulging) of the side steel plates. Such a protection concept is called a non-explosive reactive armour—NERA [3]. The effect of bulging results in a protection mechanism of minor effectiveness than that assured by the ERA armours, but the NERA armours are considered safer and, in a given protection level, lighter.

The viscoelastic properties allow elastomers to undergo extensive deformation and, after unloading, to retract an initial shape and simultaneously absorb the mechanical energy of a loading. In the military and defence, natural and synthetic rubber compounds are developed to withstand extreme operating environments and specific demands characteristic for combat zones, e.g., [4]. They are used as parts of various components present on a battlefield, e.g., dumpers, seals, fuel tanks and pillow bladders, load-bearing equipment, anti-terrorism barriers, anti-explosive absorbers and as a component of armours. Despite the considerable advancement and versatility of numerous synthetic rubbers available on the market, natural rubbers still have an important industrial and defence application position. IA discussion on the performance of a NERA rubber composite armour and an ERA armour presented in [5] was concluded that the NERA provided an acceptable protection level compared to the tested ERA and was considered environmentally safer. The ballistic resistance of laminated composite armours with various interlayers was analysed based on experimental and numerical modelling [6]. The interactions between reactive armour and shape-charge threats were described in [7], employing an analytical model. Further, the experimental investigation evaluated the influence of the target obliquity and composition on the armour protection effectiveness [8]. Whereas conclusions on the KEPs fracturing resulted from contact with an accelerated plate may be found in [9,10,11], where the residual velocity and geometry of fragments was determined after the experimental investigation. The works also discuss the target performance’s dependence on projectile length, velocity, momentum, and kinetic energy changes. It was concluded that the plate’s thickness, speed, and direction of its movement play the most crucial role in defeating KE strikers.

The presented study focuses on the protection concepts counteracting KE projectiles. Two different types of armours are tested against the down-scaled KEPs made with a tungsten heavy alloy (WHA) of the grade Y925. The effects of impacts on the homogenous semi-infinite rolled homogenous armour (RHA) steel plates are presented and then compared with shots to a steel-elastomer laminated armour. The laminates are composed of the RHA steel plates that sandwich the rubber layer, assigned to a shot at the 60° Nato angle [12,13]. The paper’s objective is to present the performance of a NERA armour with the rubber layer applied against KEP projectiles. The effect of bulging used to defeat the kinetic threats is analysed experimentally and numerically. The FEM simulation provides a detailed analysis of the threat/target interactions resulting in a description of the mechanism due to which the KE projectiles are mitigated. The study shows that the NERA protection may be an effective protection solution that can cause the fracturing of long-rod projectiles, significantly reducing their piercing potential.

## 2. Ballistic Impact Tests

Due to their exceptionally high density and strength, the tungsten heavy alloys (WHA) are important constituents for the military and armament industry [14]. Consolidation of the raw tungsten, which has a density of 19.3 g/cc, requires high temperatures processing, so industrial WHA alloys are manufactured mainly by powder sintering. The addition of alloying elements (usually nickel, iron and cobalt of the content lower than 10% of the total composition) ameliorates the ductility and machinability of the raw tungsten. During sintering, melted additives form a liquid, which in the final process stage establishes a two-phase composite structure with large tungsten grains distinctly visible inside the matrix, Figure 1a.

In the current investigation, the long-rod strikers are manufactured from the tungsten alloy of the Y925 grade produced by Kennametal Mistelgau. Large tungsten grains establish 92.5% of the total elements contain, Figure 1a. The average grain size is measured as 25 ± 3 µm. Some properties of the Y925 WHA alloy, quoted after the producer datasheet [15], are collected in Table 1. The WHA of the Y925 grade has been available for military applications for a long time; therefore, in the already published expertise, a description of its material and mechanical performance may be easily found, e.g., [16,17]. In Figure 2a,b, the influence of the strain rate and temperature increase on the plasticity of the WHA Y925 grade is presented. Under a quasi-static compressive loading, the elastic range of the stress–strain curve describing the Y925 WHA grade reached 1300 MPa, the Young modulus was established as 347 GPa, and an average ultimate tensile stress (UTS) was measured as 1350 MPa.

High costs of full-scale ballistic tests inspired experimental works with laboratory, down-scaled models [18,19]. The dimensions are reduced, but the geometrical proportion and material properties of a tested threat/target configuration are the same as in the full-scale test. Additionally, well-equipped academic laboratories often provide more detailed measurements and observations than it is possible at open-field proving-ground sites, where full-scale models are tested. In the experimental campaign discussed here, the long-rod projectiles have a diameter of 4 mm and a length of 80 mm, resulting in the length/diameter (L/D) ratio of 20, Figure 1c. A thin round steel cup stabilises the rod during a flight which does not disturb its effectiveness during penetration of targets. Inside the barrel, the projectile is supported by a plastic sabot (of the mass is 19.4 g), which falls apart before the projectile reaches the light barrier. The total mass of the KE projectile is 19.1 g and with a sabot, 38.5 g.

In the experimental campaign, two types of tests have been performed. The first kind concerns so-called ‘depth of penetration’ (DOP) tests, in which the length of the crater left by a threat is measured to evaluate its piercing potential. In the ballistic test, the KE projectiles are fired against the rolled homogenous armour steel RHA steel blocks of 200 mm × 200 mm × 500 mm. The proportion between the projectiles and targets dimensions allows the targets to be considered as semi-infinitive. Armour steel of the RHA grade is available from most armour steel producers. Due to its standardised properties and chemical composition (e.g., [20]), it is often considered the reference steel in protection systems. This martensitic steel has a hardness of 380 HB and a UTS of 1200 MPa. The curves from the quasi-static and dynamic (4800 s^−1^) compression tests of the RHA grade steel are given in Figure 2c [21,22].

The DOP test allows verifying the depth, which the KEP projectiles penetrate in a monolithic shield. An armour made of homogeneous material is not applicable from the practical point because it would need to be very thick to stop projectiles. Thus, various materials are assigned together to benefit from their properties and mutual interactions to optimise energy absorption mechanisms. Therefore, the second type of targets consists of two RHA steel plates that sandwich a rubber interlayer. In the down-scaled configuration, the targets have 40 mm × 300 mm and are made with 4 mm thick steel plates and 10 or 15 mm thick natural rubber layer, Figure 3a. In the tensile test, the rubber of a hardness 70 Shore (scale A) reached 25 MPa at the test elongation of 700%. According to the classical Yeoh study, the set of curves characterising a natural rubber [23] is given in Figure 2d. No adhesive was added between the layers of the target to enable the rubber a free expansion. 

To the shots, the NERA targets are inclined at 60° NATO angle, Figure 3a. At this angle, the line of sign (LOS) increases almost two times, extending the time of interactions between the threat and target. Additionally, an impact at an angle enhances an asymmetry effect that disturbs a long and slender KE projectile.

As the launcher, a powder gun of cal. 25 is used (marked as 1, in Figure 3). In front of a catch box (3) where the targets are inserted, the light barrier (2) is located to measure the projectile’s impact velocity. Each shot, in which the target perforation is expected, is recorded by a multi-anode flash X-ray radiography (4), which follows the projectile trajectory from the side view. Contrary to the conventional X-ray radiography (e.g., [24,25]), the multi-anode X-ray apparatus depicts the state of the threat/target configuration in a few separated frames taken with a few microsecond delay. The advanced X-ray technique is necessary during the KEP ballistic tests because for these high-speed and destructive tests, it is challenging to extract residual fragments of projectiles for further analysis. Besides being a piece of experimental evidence, X-ray images are applied for establishing the velocity of residual fragments and debris.

## 3. Results of the Ballistic Tests

The results of the depth of penetration tests of the KEPs with the diameter 4 mm and length of 60 mm, 80 mm and 96 mm against the RHA steel are collected in Figure 4, which presents post-test longitudinal sections of the penetrated steel blocks. The projectiles were accelerated to a velocity higher than 1500 m/s, and accordingly, to their increased length, the measured DOP values increased, too. The projectile with a length of 60 mm buried a crater of depth 54.6 mm, whereas the projectiles 80 mm and 96 mm long caused penetration of the 63.2 mm and 69.5 mm long target structure. In the second shots round, the craters that remained after the projectiles 60 mm, 80 mm and 96 mm long have the depths of features similar to the first ones, i.e., 52.1 mm, 63.9 mm and 67.9 mm, respectively.

During the penetration, only small parts of the projectiles are eroded. The measured depth of craters shows that the strikers penetrated the RHA steel on the length, which reached even 90% of their initial length (the 60 mm long projectile buried a 55 mm long crater). With the increasing length of projectiles, the remaining craters’ length increased too, but a proportion between the DOP and the projectile length decreased to 78% and 74% for the projectiles 80 and 96 mm long. Most of the rod structures remained inside the craters filling them tightly. The residual rods underwent a substantial deformation, cracks and thermal transformations during a high-speed penetration process, Figure 5. Their front part, initially blunt, deformed, obtaining a characteristic “mushroom cap” shape, Figure 5c, resulted from a material plastic flowing on the interface between the rod and the target.

Contrary to the above-presented protection mechanism in which the projectile’s energy is absorbed during the target penetration by their deformation and erosion, a NERA armour takes advantage of a mechanism due to which the projectile is deflected and fragmented. The targets comprise the 4 mm thick RHA steel plates and the 5, 10, 15 and 2 × 10 mm thick rubber interlayer. The reference configuration aimed to emphasise the effects of the rubber is composed of the steel plates, between which there is a 15 mm air gap, not the rubber, see Figure 6.

The targets are assigned at the NATO angle of 60°, which increases the LOS to 38 and 48 mm for the configurations with 10 and 15 mm thick rubber layer, and to 58 mm for the composite with a double 10 mm thick rubber layer. The downscaled targets are characterised by the aerial mass of 65 kg/m^2^, which is eight times less than the aerial mass of a 65 mm thick RHA plate (495 kg/m^2^) required to stop an 80 mm long KEP. At the moment of impact, the KE projectiles with a diameter of 4 mm and a length of 80 mm are accelerated to an impact velocity higher than 1500 m/s (their kinetic energy is then about 2200 J).

Looking at the results collected in Figure 6, a conclusion may be drawn that the drop of velocity read for the residual rod fragments cannot be considered the primary defeat mechanism. A decrease of the striker velocity is not significantly high—only about 5% of the projectiles’ initial energy is absorbed by the perforation of the targets (independently of whether the rubber interlayer was applied or not). Nevertheless, the X-ray images present the destabilised and fragmented rods behind the perforated laminates. The frames also captured the behaviour of the side steel plates, which deformed together with the deforming rubber—it is experimental evidence of the so-called ‘bulging effect’. Therefore, not the velocity reduction, but the fragmentation of the KE projectiles may be here considered the protection mechanism.

There is no considerable difference in the 10 mm and 15 mm thick rubber layer effect on the projectile fragmentation. However, it is observed that the rubber layer should not be too thin. The 5 mm thick rubber in this test configuration has not allowed for a sufficiently long time of interactions, and only the rear part of the projectile is broken off from its main length. Among the tested steel-elastomer laminates, the strongest rod fragmentation occurs when a 2 mm × 10 mm thick rubber is inserted between the steel plates. Then, the rod cracks into four pieces.

For high-speed and destructive terminal ballistics experiments, in situ observation techniques are often challenging in the application. A high-speed camera record could not be used in the performed ballistic test, and the X-ray imaging provided only a few frames for each shot. Hence, a numerical simulation may be an appropriate tool resulting in a complimentary analysis of the phenomena observed experimentally. The following section discusses the numerical results to add a more profound insight into the threat-target interactions recorded in the experiment. 

## 4. Numerical Modelling of the NERA Laminates upon KEP Impacts 

The simulation of the NERA laminated armour is performed employing the Ls-Dyna explicit code v. 9.0.1, [26], where the threat-target configuration is represented by deformable Lagrangian solids meshed by reduced integration 8-node solid elements with stiffness-based hourglass control. An optimally fine mesh is imposed in the rod and the plates to allow a fracture initiation [27]. The KE projectile is meshed by 0.1 mm × 0.1 mm × 0.1 mm elements, which for the rod of the diameter and length 4 mm and 80 mm results in 1.192.000 elements. The front and back plates (4 mm × 40 mm × 200 mm) are meshed by elements with 0.1 mm × 0.1 mm × 0.1 mm in the impact zone (4 mm × 20 mm × 40 mm), but a coarser meshing is applied outside this central zone, see Figure 7. The plates are meshed then by 882.000 elements. Elements of the size 0.2 mm × 0.2 mm × 0.2 mm are imposed regularly, meshing the rubber—thus, 535.464 elements discretised a 15 mm thick layer. Contact between the rod and target is applied by the option * ERODING_SURFACE_TO_SURFACE, due to which elements deletion is possible. To repeat the experimental boundary conditions, the impact velocity is assigned to the projectile along the impact direction, and the target sides are fully clamped, see Figure 7. The steel plates and the rubber layer are in contact due to the * AUTOMATIC_SURFACE_TO_SURFACE function representing the non-bonded target layers [28]. The frictionless condition is applied between all modelled parts because of a lack of accurate description [29,30].

The parameters of the flow and fracture model for the RHA steel are based on [21], which allows an application of the Johnson-Cook (JC) flow and fracture model, [31,32], implemented by the card * MAT_15 and the Gruneisen equation * EOS_GRUNEISEN, [33], see Table 2. The JC model considers phenomenological observations of the steel and metals behaviour, i.e., influence of the strain, strain rate and temperature, as well as of the stress triaxiality in the fracture model. The formulation of the Ogden model (* MAT_77) is chosen to model the hyper-elastic rubber behaviour. A classical set of the model parameters proposed by Yeoh [23] is used, Table 3.A more detailed discussion on the modelling approach may be found in [12,13].

The set of the material model constants collected in Table 4, which describes the plastic hardening of the Y925 WHA according to the Johnson–Cook flow model was found in the references [16,17], see Figure 2a,b. However, this material characterisation does not account for the description of the WHA fracture. In consequence, the * MAT_SIMPLIFIED_JC (* MAT_98) is chosen as the constitutive model. The * MAT_98 is one of several modelling approaches available in Ls-Dyna based on the Johnson–Cook flow model [23]. This material function does not account for temperature, and it is decoupled from the model of fracture proposed by Johnson and Cook [23,34]. The Ls-Dyna code has additional erosion criteria, which implement a simple, one-value fracture criterion in calculations. When in an element, the chosen threshold is reached, and the element is removed (eroded). Ls-Dyna provides a choice among several options (e.g., the maximum shear strain at failure, equivalent strain at failure, minimum pressure at failure, etc.). In the current calculations, the * MAT_ADD_EROSION function with the minimum pressure at failure, MNPRES, is considered the fracture criterion. The pressure, i.e., hydrostatic stress, is the average of any stress tensor’s three normal stress components. In the Ls-Dyna package, the hydrostatic pressure is considered positive in compression. To obtain the experimentally observed rod fragmentation, the minimal pressure at failure equals −650 MPa. This value was chosen based on a parametric study of a number of cracks in the rod [12]. High strength metals are usually sensitive to overloading in the tensile regime, in which they crack more easily than under compression [35,36]. This phenomenological observation, not contradictory with the physics of high-strength metals, justifies applying the MNPRES threshold as a simplified fracture criterion. Already published examples confirm that FEM simulations of high-strength cores of small-calibre rounds, which account for an added erosion function, result in acceptable modelling of their breakage [36,37,38].

Based on the numerical simulation, some distinct stages may be distinguished alongside the passage of the KE projectile through the target laminate. The penetration of the laminate with the 15 mm thick rubber interlayer is shown in dependence to the modelled reduction of the KEP velocity, Figure 8a. The first phase ends up at 10 µs (Figure 8a–picture 1), which is the moment when the rod perforated the front steel plate and started to penetrate the interlayer. The curve changes its slope when the projectile’s nose touches the back plate (22.5 µs, picture2). The next phase ends up about 40 µs when the back plate is fully perforated and material debris is ejected (picture 3). In the ballistic tests, an irregular plug shape may be seen on the X-ray images (Figure 6); the simulation does not result in debris of this size (picture 4). The rod is entirely fragmented at 62.5 µs. Additionally, when the phase of a constant exit velocity starts, it may be concluded that the whole projectile left the target (80 µs, picture 5). It cannot be confirmed by the experiment if the projectile’s velocity reduction consists of such phases. Still, it may be concluded that each stage of penetration is reflected in the change of the curve’s character, which is well presented by the simulation.

The evolution of the pressure in the cross-section of the long-rod before the cracks initiates and afterwards is shown in Figure 8b. According to the simulation, the rupture begins on the upper rod periphery when it perforates the back plate. Due to fine meshing (elements size 0.1 mm^3^), the fracturing does not require an extensive element erosion—the modelled fracture contains only a single elements layer. The rear projectile part is stabilised inside the elastomeric interlayer, which enfolds it tightly. At the same time, the front part of the rod (already outside the target) moves slightly downwards. On the presented hydrostatic stress maps, it may be observed that locally, the projectile is stretched on its upper periphery. In contrast, its bottom side is slightly compressed—which globally may be considered bending of the rod. Thus, the WHA rod, characterised by high strength and hardness but relatively low ductility (Table 1), starts to crack in the tensile regime. A similar conclusion was drawn in the studies [9,10], where it was stated that rod fractures ‘seem to be due to bending’. The experimental and numerical investigations show that a deflection and bending of long-rod projectiles are a beneficial protective mechanism that strengthens long-rods’ tendency to break-up.

For further validation of the modelling approach, two other impact scenarios (without the rubber and with the 10 mm thick rubber layer) are modelled numerically with the same conditions and material models as in that one analysed above. The calculations show that the rod’s bending does not occur without the rubber interlayer, and neither does its fracture, see Figure 9a. The deformation which the projectile undergoes during the penetration of two steel plates is insufficient to cause bending of the striker and, thus, to activate the fracture condition of the minimum pressure. Similarly to the simulation of the configuration with a 15 thick rubber layer, Figure 9c, the modelling of the NERA with the thinner rubber layer results also in a fragmentation of the rod, Figure 9b. This effect is known as ‘bulging’.

## 5. Discussion on the ‘Bulging Effect’

As depicted by the X-ray images (Figure 6) and modelled numerically (Figure 8 and Figure 9), the rubber changes its shape, deforming rapidly upon the impacts of the KE projectiles. A high-pressure impulse is conveyed to the laminate by the striker with high kinetic energy at the moment of impact. Initially, the rubber interlayer is compressed under the blunt projectile nose, but then it expands, affecting the steel plates, which deform along with the deforming rubber, pushed by the long striker. The rods are stabilised by the elastomeric interlayer and strained between the side steel plates deforming in the opposite directions. In consequence, the KEPs are bent, and later, cracks initiate in the stretched zones. The side plates were initially parallel, but they become distinctly bent after the perforation, see Figure 10.

This deformation and the resulted fragmentation of the KEPs is described as the ‘bulging effect’ [39]. In the configuration without the rubber, bulging of steel plates is not observed. The X-ray images and the simulation show that the KE penetrator is not damaged or disturbed after the perforation. In non-explosive reactive armours, the choice of an elastomer as the interlayer is not accidental. Among other materials, elastomers have the highest expansion coefficient and a relatively low Young modulus, promoting a fast elastic deformation and retrieving the initial shape (Figure 11 based on [40]).

In Figure 10, the photographs of the steel plates sandwiching a 15 mm thick rubber are shown. The wavy shape of both plates may be easily noticed. The plates are about 8 mm (the back plate) and 12 mm (the front plate) higher than when they were flat and undeformed. The exit hole in the rear plate is longer (34.2 mm × 9.4 mm) than the entry hole (21.3 mm × 9.6 mm). While leaving the target, the projectile’s rear part has already deviated from the initial trajectory. The front and back halves of the rods acquire different impulses since the time of contact with the plates was different. In Figure 6, the flash X-ray images visualised the cracks in the middle and the rear rods’ parts. The armour acted longer on the back rod part than on its front, which flew quickly beyond the layers.

Due to the numerical simulation, detailed observations of the plates and rubber behaviour are possible, which helps to analyse the mechanism of KEPs defeating. Based on the example with the 15 mm thick rubber, Figure 12 shows the changes in the rubber layer thickness during the projectile passage and Figure 13 and Figure 14 analyse the deformation of the side steel plates.

While the long-rod projectiles traverse the rubber, the rubber expands quickly, increasing its thickness to 16.5 mm—the maximal value for the analysed cross-section reached when the whole rubber layer is perforated and the KEP hits the back plates, Figure 12. Then, the coating starts to shrink, and it is thinning to 11 mm obtained when the rod is tightly enfolded by the whole rubber layer, at 55 µs. Figure 8b allows noticing that this is the time instant that proceeds the moment of the crack initiation in the rod. Afterwards, the rubber almost retreats its initial thickness. The deformation of the rubber is followed by the plates’ deformation, which is discussed in the next paragraph.

The impact direction in the simulation is along the global direction Y. Still, the target is inclined at 60°, therefore to visualise better the deformation of the side plates, they are presented in the maps of the resultant displacement, see Figure 13.

Despite not accurately identified values of the deformation of the plate, the simulation generally predicts correctly modelled shapes of the deformed plates. On the cross-section of the perforated target, six points are chosen on the front (1–3) and back faces (4–6) of the steel plates. The dependences of the resultant displacement in time indicate that both plates start to deform already at 20 µs of the penetration process when the rubber begins to expand (Figure 8). The projectile perforated the first plate and is in the middle of the rubber layer, the expanding of which starts to affect the back plate. Point 1 starts to move a bit earlier because of the proximity of the entry hole of the perforation channel. Its resultant displacement (1.6 mm) is almost as high as its value read for point 2 (1.8 mm), located in the first plate’s most bent zone. Point 2 reaches the plateau of its displacement in its maximum value when the projectile leaves the target (close to 80 µs). The bottom plate deforms more than the front plate. Points 4 and 5, which are beneath the rod nose, are affected more than point 6 above it (the resultant displacement of which is about 1 mm). Point 5, close to the crater of the perforation channel, reaches 3 mm of the resultant displacement. This maximum value is obtained when the rod is no longer in contact with that side of the exit hole.

Figure 13, based on all simulated configurations with different rubber thickness (cf. Figure 9), compares the resultant displacement curves read for distinctly displaced points, i.e., point 2 in the front plates and point 5 in the back plates. The comparison shows that in the laminate without the rubber, the plates do not form the characteristic bulge—the displacement of the points is not significant. Comparing with the other configurations, the double rubber laminate is most distinctly deformed—the points in the bottom and front plates moved more than their counterparts from the remained configurations. This numerical result complies with the experimental observation (cf. Figure 6) that a more substantial plate’s deformation causes a more significant rod fragmentation. It may be then concluded that the numerically obtained comparison of the plates’ displacement illustrates well the experimentally observed effect of bulging. 

## 6. Conclusions

The presented above experimental and numerical investigation is the basis of a discussion on defeating the kinetic-energy penetrators. The effects of KEP impacts against the non-explosive reactive armour with the rubber layer is compared with the results of shots against homogenous steel blocks. The ballistic impact test concerns the down-scaled KEPs (the rod diameter D = 4 mm) accelerated above 1500 m/s. The flash X-ray images taken during the experimental campaign show the threat/target performance. The numerical simulation adds a complementary analysis to the experimental observations.

The performed DOP test proves that to stop the KEPs; the RHA steel block should be of thickness close to the projectile length, making this solution too thick and too heavy for practical applications. Under an impact, the rubber inside the NERA armour deforms, evoking a deformation (bulging) of the front and back steel plates. While the rubber rebounds, it enfolds and stabilises the rod between the side steel plates deformed in the opposite directions—in consequence, the rod starts to bend. The KEP projectiles are made with a hard, resistant, not so ductile tungsten alloy. Therefore, when these long and slender rods are bent, they are more prone to fracture strained at tension. The numerical simulation describes a crack initiation in the bent KE penetrator destabilised by an asymmetric contact with the deforming plates. It is then numerically proven that a change of shape of the rubber causes deformation of the side plates, which is considered the primary defeat mechanism in the discussed protection concept. The mutual performance of the interacting armour layers results in the efficient mitigation of the piercing potential of the KE projectiles.

## Figures and Tables

**Figure 1 materials-14-03334-f001:**
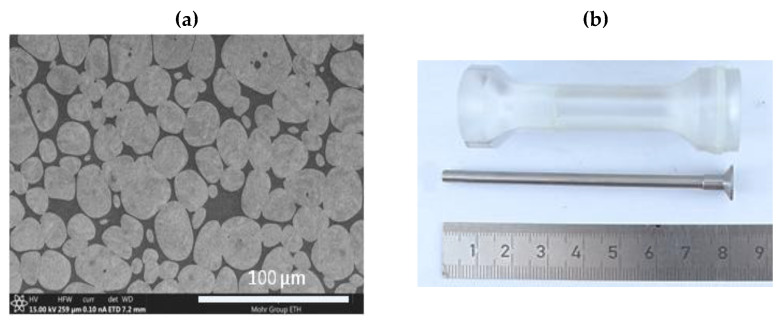
(**a**) Microstructure of the Y925 WHA alloy and (**b**) the down-scaled KE projectiles, here with the length to diameter ratio L/D = 20.

**Figure 2 materials-14-03334-f002:**
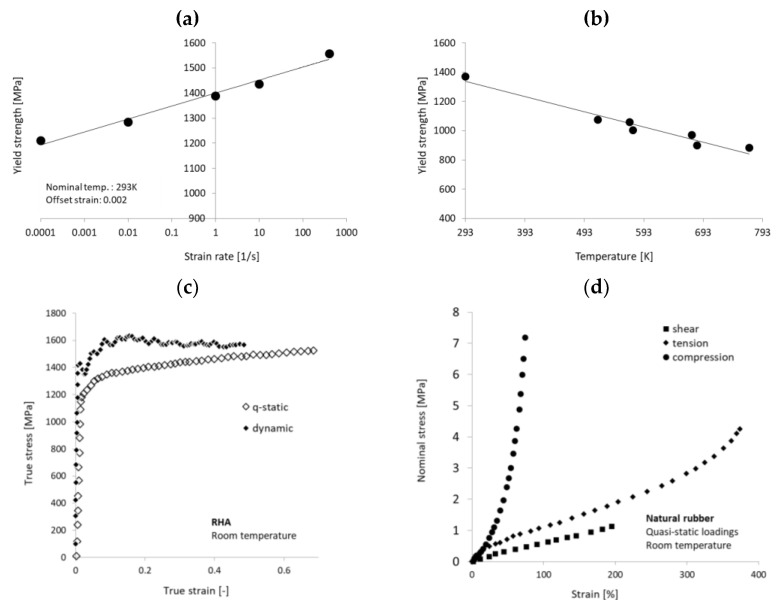
Tungsten heavy alloy of the grade Y925: (**a**) strain rate and (**b**) temperature sensitivity]. (**c**) Quasi-static and dynamic flow curves of an RHA steel, (**d**) quasi-static curves of natural rubber. Graphes based on the data collected in [16,17,21,22,23]

**Figure 3 materials-14-03334-f003:**
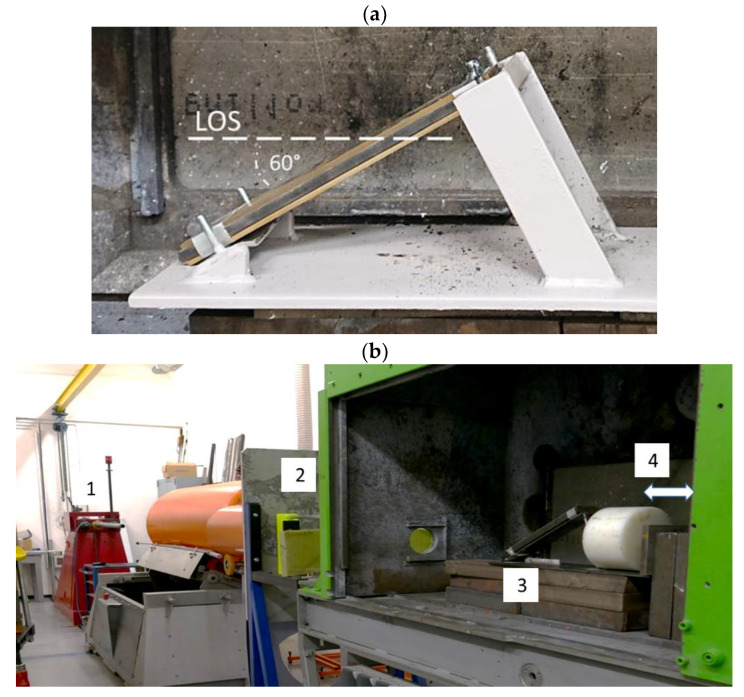
(**a**) NERA armour inclined at 60° NATO angle. (**b**) Experimental stand (explanations in the text).

**Figure 4 materials-14-03334-f004:**
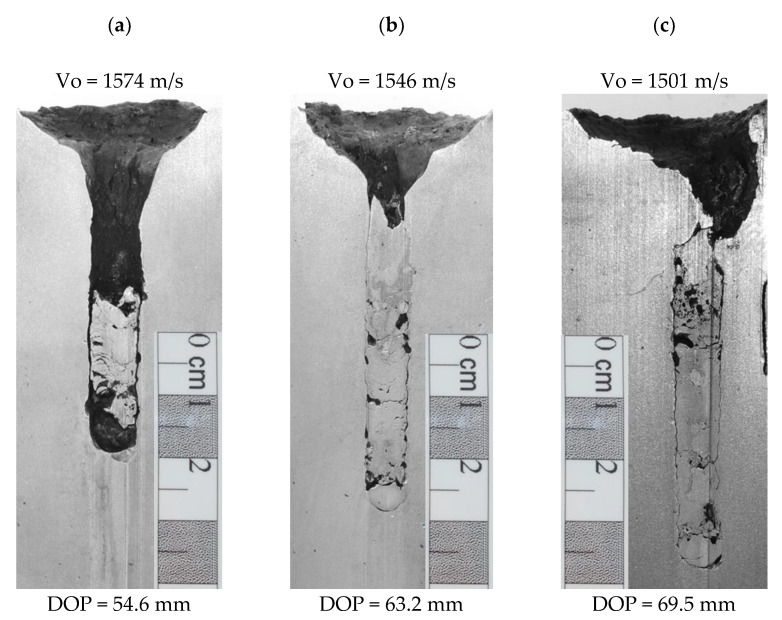
DOP tests against blocks of the RHA steel. The projectiles of diameter 4 mm and length to diameter (L/D) ratio: (**a**) 15, (**b**) 20 and (**c**) 24.

**Figure 5 materials-14-03334-f005:**
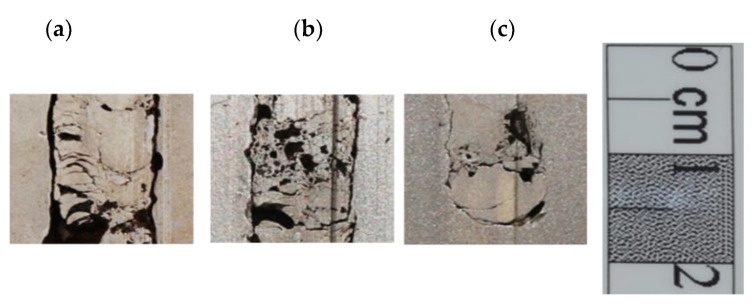
Enlarged fragments of the residual rods inside the penetration channels, which initially were: (**a**) 60 mm, (**b**) and (**c**) 96 mm long.

**Figure 6 materials-14-03334-f006:**
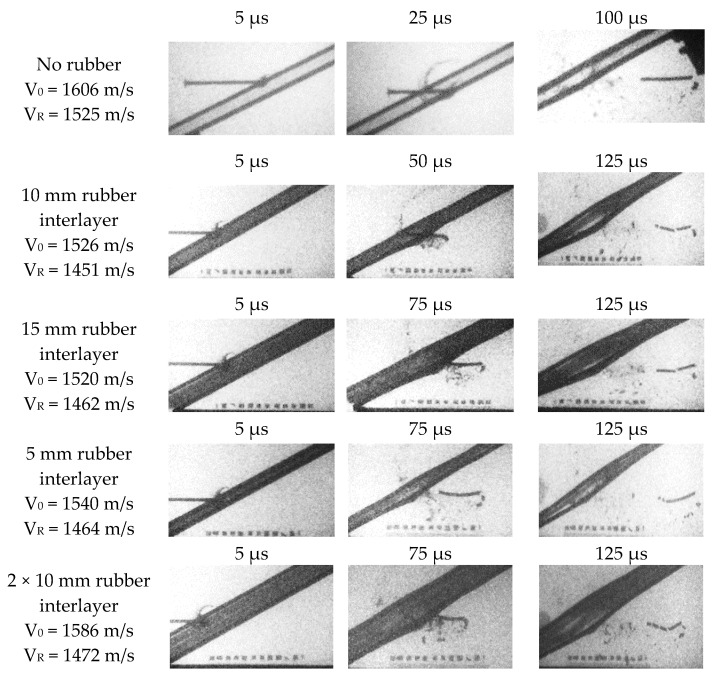
Ballistic shots registered on the flash X-ray images (V_0_—initial velocity, V_R_—residual velocity).

**Figure 7 materials-14-03334-f007:**
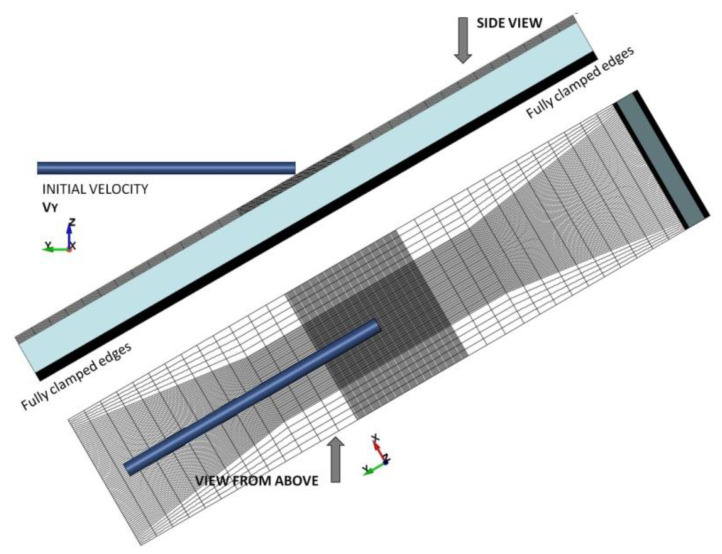
Numerical representation of the steel-elastomer passive armour.

**Figure 8 materials-14-03334-f008:**
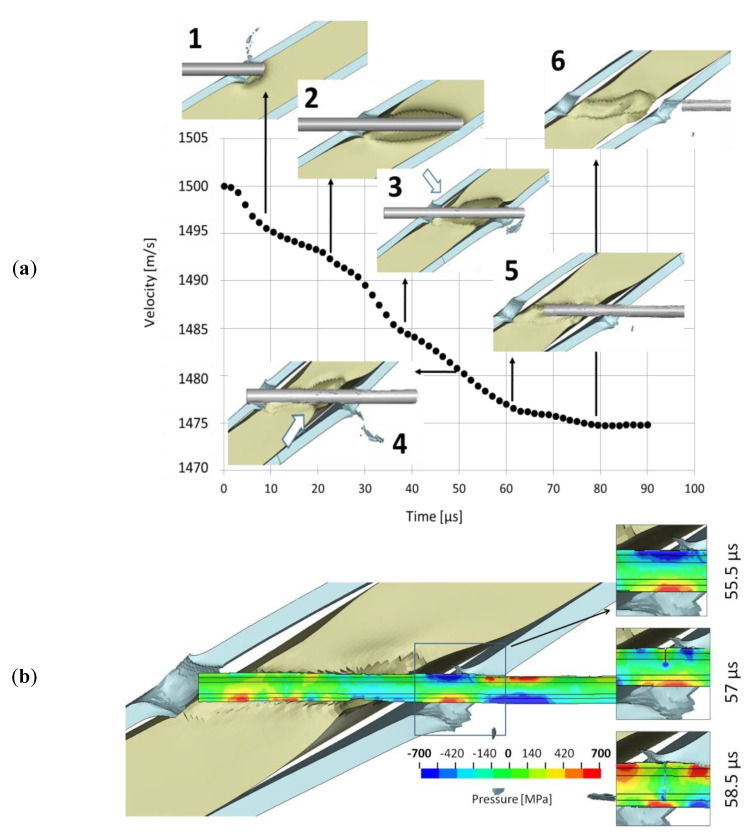
Long-rod projectile and (**a**) drop of its impact velocity, (**b**) fracture initiation (the hydrostatic pressure positive in compression).

**Figure 9 materials-14-03334-f009:**
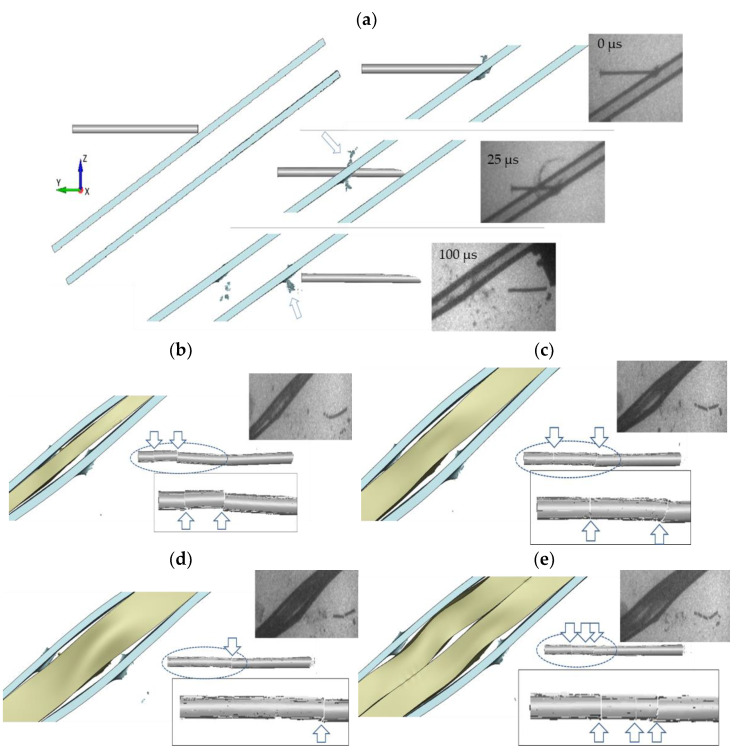
Numerical and experimental results for the configurations: (**a**) without the rubber interlayer, (**b**) with a 5 mm, (**c**) 10 mm, (**d**) 15 mm, and (**e**) 2 × 10 mm thick rubber layer.

**Figure 10 materials-14-03334-f010:**
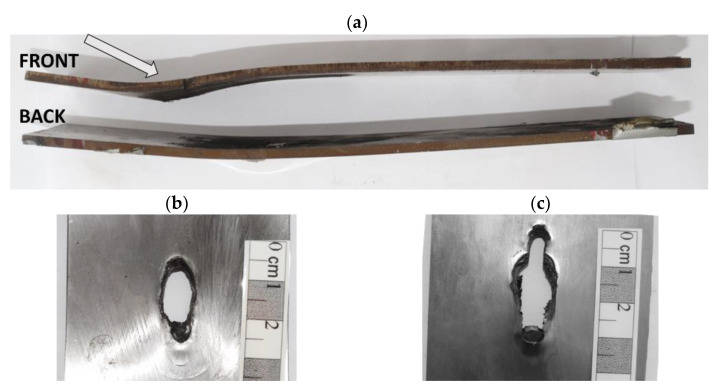
(**a**) Deformation of the side steel plates. Entry (**b**) and exit holes (**c**).

**Figure 11 materials-14-03334-f011:**
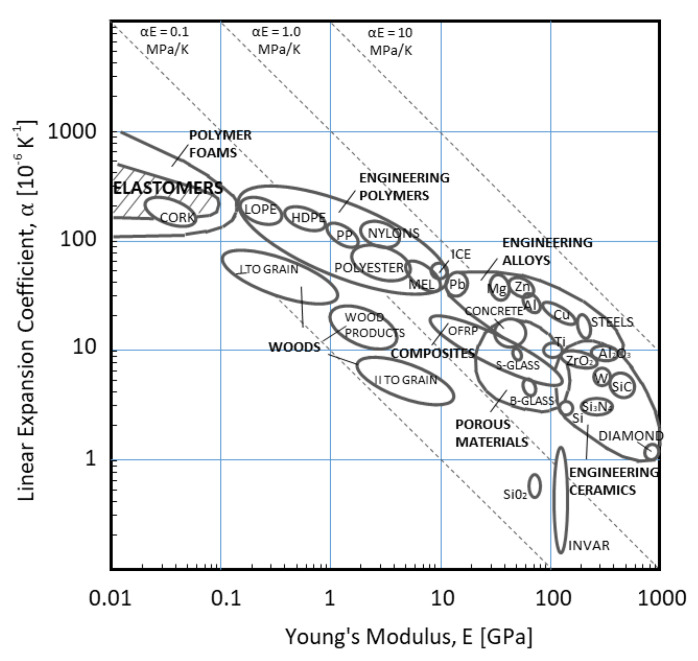
Expansion coefficient for different materials, based on [40].

**Figure 12 materials-14-03334-f012:**
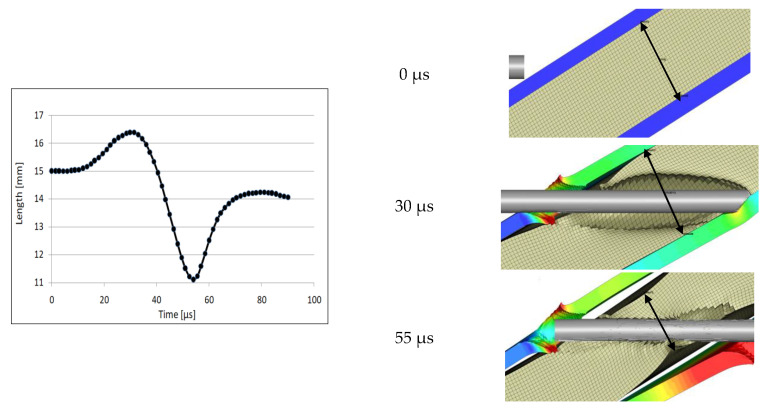
Changes in the rubber thickness during the target penetration.

**Figure 13 materials-14-03334-f013:**
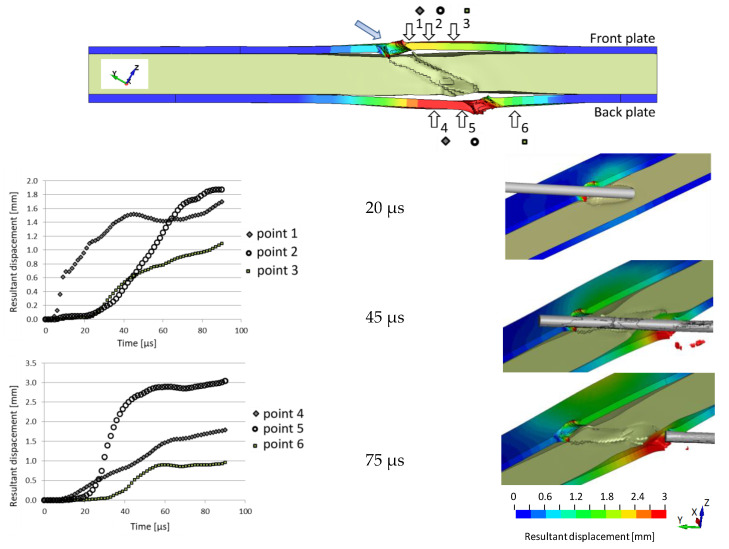
The resultant displacement of the side steel plates.

**Figure 14 materials-14-03334-f014:**
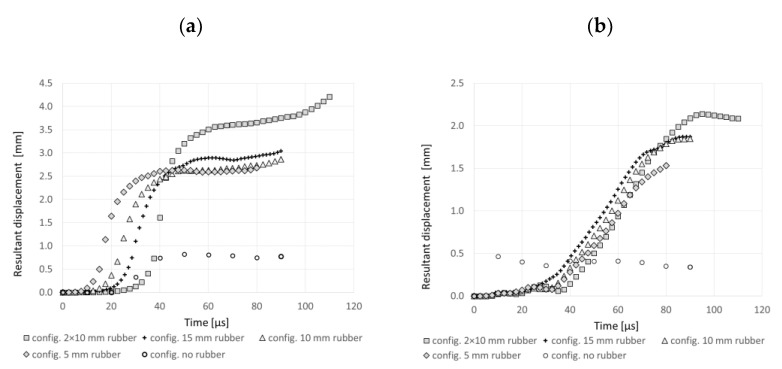
The resultant displacement for (**a**) point 5 (on the back plate) and (**b**) point 2 (on the front plate) in the simulation of the configurations with different rubber thicknesses.

**Table 1 materials-14-03334-t001:** Same material parameters of the tested WHA grade.

**WHA grade Y925**			
Tungsten	Nickel	Iron	Cobalt
92.5%	rest		
Density		17.7 g/cm^3^	
Hardness		400 HV10	
Tensile Strength		1350 MPa	
Yield strength		1300 MPa	
Elongation (A5)		8%	

**Table 2 materials-14-03334-t002:** Modelling parameters the RHA steel, [21,33].

RHA steel					
JC Flow Model		JC Fracture Model		Gruneisen EOS	
A (MPa)	1193	D_1_	0.21	C	4570
B	500	D_2_	7.21	S1	1.49
C	0.0043	D_3_	−5.44	S2	0
n	0.67			S3	0
m	1.17			γ_0_	1.16
				ρ (g/cm^3^)	7.85

**Table 3 materials-14-03334-t003:** Modelling parameters for the natural rubber, [23].

Natural Rubber			
Ogden Model			
$\alpha_{1}$	1.3	$\mu_{1}$	0.618
$\alpha_{2}$	5.0	$\mu_{2}$	0.00118
$\alpha_{3}$	−2.0	$\mu_{3}$	−0.00981

**Table 4 materials-14-03334-t004:** Modelling parameters for the WHA Y925 [16,17].

WHA grade Y925 JC Flow Model	
A (MPa)	1258 ± 97
B	0.092 ± 0.014
C	0.0013
n	10.014 ± 0.0003
m	0.940 ± 0.007

## Data Availability

Data Sharing is not applicable.

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
