# Peer review of "Experimental and Numerical Study on a Non-Explosive Reactive Armour with the Rubber Interlayer Applied against Kinetic-Energy Penetrators—The ‘Bulging Effect’ Analysis"

_materials, 2021, doi:10.3390/ma14123334_

Round 1
Reviewer 1 Report
The authors studied the presented armour with rubber interlayer and provided a explanation of the bulging effect in the protective behaviors through numerical simulation and experiments. There are some places to be revised as following. The minor revision is advised.
- In figure 4 and 5, please add the scale label.
- In figure 6 and 9, the thickness of rubber interlayer seems not significant to the protection mechanism from the numerical and experimental results. The influence of rubber thickness is not illustrated clearly in this work. Please add some data or explanation to verify the point about thickness.
- In figure 8(a), the position of “5” and “6” may be marked incorrectly.
- In figure 12, the panels should be better labeled to read easily, and the scatter figure on the left should use an easily distinguishable sign.
- Figure 13 shows the changes in the rubber thickness during the target penetration of 15mm thickness rubber interlayer. If the steel plates displacements of no rubber, 10mm rubber, 15mm rubber interlayer can be shown, it may demonstrate the bulging effect of armour.
Author Response
I would like to thank the Reviewers for the time and effort they devoted to reading the paper. All the comments are accounted for in the revised manuscript. The English were proofread. The green color indicates the changes and added fragments in the text.
Thank you for this positive feedback and a careful reading of the paper.
Comments 1, 3 and 4
Figures 4, 5, 8, 12 are modified as advised.
Comment 2
Figure 6 is complemented by more experimental results. Accordingly, Figure 9 is completed by the numerical simulation of the added experimental configurations. As the Reviewer noticed, there is no considerable difference in the 10 and 15 mm thick rubber layer effect on the projectile fragmentation. However, in the experiment, it is observed that the rubber cannot be too thin – the 5 mm thick rubber in this test configuration has not allowed for a sufficiently long time of interactions. Only the rear part of the projectile is broken off from its main length. Among the tested configurations, the strongest rod fragmentation occurs when a 2 x 10 mm thick rubber is inserted between the steel plates. Then the rod cracks into four pieces.Comment 5
Following the suggestion, Figure 13, based on all simulated configurations with different rubber thickness (cf. Figure 9), compares the resultant displacement curves read for distinctly displaced points, i.e., point 2 in the front plates and point 5 in the back plates. The comparison shows that in the laminate without the rubber, the plates do not form the characteristic bulge – the dis-placement of the points is not significant. Comparing with the other configurations, the double rubber laminate is most distinctly deformed – the points in the bottom and front plates moved more than their counterparts from the remained configurations. This numerical result complies with the experimental observation (cf. Figure 6) that a more substantial plate's deformation causes a more significant rod fragmentation. It may be then concluded that the numerically obtained comparison of the plates' displacement illustrates well the experimentally observed effect of bulging.
Both above comments are added to the manuscript.

Reviewer 2 Report
The authors conducted the experimental and computational investigation of the dynamic penetration process of a non-explosive reactive armour against kinetic-energy penetrators. The research is within the scope of the journal with detail the experimental and computational description, results, and discussions. The research work is new and were designed, and the computational results can validate the experimental results.
The paper can be accepted for publication while a minor format change change be made as follow: The figures and diagrams in the Conclusion sections can be move to the results and discussion section.
Author Response
Lots of thanks for this positive comment. Your remark is accounted for in the updated manuscript.
Reviewer 3 Report
The Author described studies concerning on application of rubber in a non-explosive reactive armor against kinetic energy penetrators. An investigation of these materials is very important due to provide protection soldiers serving in tanks or armored transporters divisions. Below, several aspects have mentioned, which should be corrected and some doubts should be explained.
- The Abstract should deeply modified due to this part of manuscript should contain the most important results from the text.
- The Introduction should be modified in order to highlight the motivation of studies.
- The results should be compared to some previous researches.
Generally, the Authors did excellent work. I recommend minor revision.
Author Response
I would like to thank the Reviewers for the time and effort they devoted to reading the paper. All the comments are accounted for in the revised manuscript. The English were proofread. The green color indicates the changes and added fragments in the text.
Thank you very much for the positive feedback – it is appreciated a lot.
Following the comments, the Abstract is modified:
The study concerns a protection system applied against kinetic-energy penetrators (KEPs) com-posed of steel plates sandwiching a rubber layer. Laminated steel-elastomer armours represent non-explosive reactive (NERA) armours that take advantage of a so-called 'bulging effect' to mitigate KEP projectiles. Upon an impact, the side steel plates deform together with the de-forming rubber interlayer. Their sudden deformation (bulging) in opposite directions disturbs long and slender KEP projectiles causing their fragmentation. The presented discussion is based on the experimental investigation, confirming that the long-rod projectiles tend to fracture into several pieces due to the armour perforation. A numerical simulation accompanies the ballistic test providing an insight into the threat/target interactions. The presented experi-mental-numerical study explains the principles of the analysed protection mechanism and proves the efficiency of the materials composition making up the laminated non-reactive pro-tection system.
And to Introduction the following explanation is added:
The paper's objective is to present the performance of a NERA armour with the rubber layer applied against KEP projectiles. The effect of bulging used to defeat the kinetic threats is analysed experimentally and numerically. The FEM simulation provides a de-tailed analysis of the threat/target interactions resulting in a description of the mechanism due to which the KE projectiles are mitigated. The study shows that the NERA protection may be an effective protection solution that can cause the fracturing of long-rod projectiles, significantly reducing their piercing potential
Finally, Section 4 is completed by the conclusion:
A similar conclusion was drawn in the studies [9-10], where it was stated that rod frac-tures' seem to be due to bending'. The experimental and numerical investigations show that a deflection and bending of long-rod projectiles are a beneficial protective mechanism that strengthens long-rods' tendency to break-up.
Reviewer 4 Report
In my opinion, the proposed experiments have been adequately conducted and the simulation is also correct. However, my concerns have to do with the approach of the experiment and some decisions related to experimental planning.
Firstly, a single shot for each experimental setup (different projectile lengths) with the RHA blocks does not provide the adequate statistical power. There is not enough information to assess whether the results can be considered representative, and the ballistic tests should have been replicated at least twice. The same is true for the NERA tests, although given the negligible effect of the interlayer on projectile velocity, it has relatively minor impact on the results.
Secondly, it is not clear how the RHA blocks test results can be compared with the results obtained from the NERA. The former can provide an indication of the minimum thickness required to stop the projectile, but no indicator is provided to evaluate the effectiveness of the NERA alternative. There is insufficient evidence to support your claim that NERA provides “a good protective efficiency against KEP strikers”. Although the word “benefits”, is used in the title, no other benefit but the reduction of weight (which is obvious) is presented in the paper, as no indicator compares the protection provided by the NERA with other alternatives.
Thirdly, the simulation could be used to evaluate untested scenarios once the simulation results have been shown to agree with the experimental results. The work could be reoriented towards a proposal on how improvements on the NERA design could influence the results (increasing the thickness or selecting alternative materials). In any case, the content of the paper is not in accordance with the title, as it just provides a comparison between simulation and tests.
Finally, the presentation of the results and discussion is a little bit confusing. I suggest that a “discussion” section should be provided independently of the numerical modelling.
Author Response
I would like to thank the Reviewers for the time and effort they devoted to reading the paper. All the comments are accounted for in the revised manuscript. The English were proofread. The green color indicates the changes and added fragments in the text.
Thank you for the constructive comments, they are taken into account, and the manuscript is accordingly completed and modified.
Firstly, a single shot for each experimental setup (different projectile lengths) with the RHA blocks does not provide the adequate statistical power. There is not enough information to assess whether the results can be considered representative, and the ballistic tests should have been replicated at least twice. The same is true for the NERA tests, although given the negligible effect of the interlayer on projectile velocity, it has relatively minor impact on the results.
I appreciate indicating this vital point - the following comment is then added then to the manuscript.
In the second shots round, the craters that remained after the projectiles 60, 80 and 96 mm long have the depths of features similar to the first ones, i.e. 52.1 mm, 63.9 mm, 67.9 mm, respectively.
The below image will not be added to the manuscript – the craters look like those already presented in Figure 4.
Secondly, it is not clear how the RHA blocks test results can be compared with the results obtained from the NERA. The former can provide an indication of the minimum thickness required to stop the projectile, but no indicator is provided to evaluate the effectiveness of the NERA alternative. There is insufficient evidence to support your claim that NERA provides “a good protective efficiency against KEP strikers”. Although the word “benefits”, is used in the title, no other benefit but the reduction of weight (which is obvious) is presented in the paper, as no indicator compares the protection provided by the NERA with other alternatives.
The Author accepts and agrees with the criticism. Thus, the title of the paper is changed:
Experimental and numerical study on a non-explosive reactive armour with the rubber interlayer applied against kinetic-energy penetrators – the bulging effect analysis
The misstated conclusive phrases are also deleted from the text. The Conclusion section contains a necessary summing up without the last paragraph.
Thirdly, the simulation could be used to evaluate untested scenarios once the simulation results have been shown to agree with the experimental results. The work could be reoriented towards a proposal on how improvements on the NERA design could influence the results (increasing the thickness or selecting alternative materials). In any case, the content of the paper is not in accordance with the title, as it just provides a comparison between simulation and tests.
The title is thus changed. Indeed, further simulations may be performed to optimise the material composition in a continuation of the project (and the project is continued).
Finally, the presentation of the results and discussion is a little bit confusing. I suggest that a “discussion” section should be provided independently of the numerical modelling.
Following the Reviewer suggestion, Section 5 Discussion on the ‘bulging effect’ is added to the manuscript.

Round 2
Reviewer 4 Report
In my opinion, the new version is more coherent than the original one. The English style is quite fine, but I suggest that it should be checked by a native speaker. Regards,